# Development of Blue Phosphorescent Pt(II) Materials Using Dibenzofuranyl Imidazole Ligands and Their Application in Organic Light-Emitting Diodes

**DOI:** 10.3390/ma16114159

**Published:** 2023-06-02

**Authors:** Hakjo Kim, Dain Cho, Haein Kim, Seung Chan Kim, Jun Yeob Lee, Youngjin Kang

**Affiliations:** 1Division of Science Education, Kangwon National University, Chuncheon 24341, Republic of Korea; tank528@naver.com (H.K.); dain0041@naver.com (D.C.); godls0126@naver.com (H.K.); 2School of Chemical Engineering, Sungkyunkwan University, Suwon 16419, Republic of Korea; ksc9650@naver.com

**Keywords:** phosphorescent organic light emitting diode, heteroleptic Pt(II) compounds, C^N chelate, ancillary ligand, time-dependent density functional theory

## Abstract

Organic light-emitting diodes (OLEDs) are energy-efficient; however, the coordinating ligand can affect their stability. Sky-blue phosphorescent Pt(II) compounds with a C^N chelate, fluorinated-**dbi** (**dbi** = [1-(2,4-diisopropyldibenzo [b,d]furan-3-yl)-2-phenyl-1*H*-imidazole]), and acetylactonate (acac) (**1**)/picolinate (pic) (**2**) ancillary ligands were synthesized. The molecular structures were characterized using various spectroscopic methods. The Pt(II) Compound **Two** exhibited a distorted square planar geometry, with several intra- and inter-molecular interactions involving C_π_⋯H/C_π_⋯C_π_ stacking. Complex **One** emitted bright sky-blue light (λ_max_ = 485 nm) with a moderate photoluminescent quantum efficiency (PLQY) of 0.37 and short decay time (6.1 µs) compared to those of **2**. Theoretical calculations suggested that the electronic transition of **1** arose from ligand(C^N)-centered π–π* transitions combined with metal-to-ligand charge-transfer (MLCT), whereas that of **2** arose from MLCT and ligand(C^N)-to-ligand(pic) charge-transfer (LLCT), with minimal contribution from C^N chelate to the lowest unoccupied molecular orbital (LUMO). Multi-layered phosphorescent OLEDs using One as a dopant and a mixed host, mCBP/CNmCBPCN, were successfully fabricated. At a 10% doping concentration of **1**, a current efficiency of 13.6 cdA^−1^ and external quantum efficiency of 8.4% at 100 cdm^−2^ were achieved. These results show that the ancillary ligand in phosphorescent Pt(II) complexes must be considered.

## 1. Introduction

Lighting accounts for 15–20% of total global electricity consumption, which corresponds to 5% of global greenhouse gas emissions. Therefore, energy-efficient lighting is required [1,2,3]. According to a report from the Swedish Energy Agency (SEA), lighting consumes approximately 23% of the total energy and releases a significant amount of carbon dioxide [4]. By 2030, the global population is expected to surge, leading to a 50% increase in lighting requirements. To curb energy consumption, the adoption of energy-efficient lighting solutions, such as light-emitting diodes (LEDs) and organic light-emitting diodes (OLEDs), is crucial. OLEDs generate light via the recombination of electrons and holes in the emissive layer, resulting in a self-emitting surface with diffused lighting properties. They are commonly compared with white LED sources and offer a more sustainable lighting option [5]. LED sources boast high intensity, long lifespans, and efficient outdoor performance but generate excessive heat and require additional components and procedures for surface lighting. Conversely, OLED lighting is manufactured in panel form and inherently acts as self-emitting surface lighting, eliminating the need for additional components [6].

Blue phosphorescence-emitting materials utilize phenylpyridine (ppy) or bipyridine (bpy) ligands and incorporate electron-withdrawing groups into the *C*-coordinated units (e.g., the phenyl ring in ppy). This shifts the emission wavelength to shorter wavelengths by stabilizing the highest occupied molecular orbital (HOMO) energy level. The structure of the dative pyridine unit can be altered by adding an electron-donating group, thereby widening the energy gap and destabilizing the LUMO. Representative examples include bis [2-(4,6-difluorophenyl)pyridinato-*C*^2^,*N*] (picolinato)iridium(III) [FIrpic] [7], tris(2′,6′-difluoro-2,3′-bipyridinato-*N*,*C*^4′^)iridium(III) [(dfpypy)_3_Ir] [8], and tris((3,5-difluoro-4-cyanophenyl)pyridine)iridium [FCNIr] [9]. In these complexes, F or CN acts as a strong electron-withdrawing unit and is introduced onto a phenyl or *C*-coordinating pyridine unit to control the HOMO energy. The constant progress in blue phosphorescent materials has highlighted the importance of selecting a suitable coordinating ligand to develop materials with high stability. This has a profound impact on both the device’s efficiency and device lifetime [10].

Among the previously reported blue homoleptic Ir(III) compounds, Ir(**dbi**)_3_, exhibits high current and external quantum efficiencies [11]. To extend the device’s lifetime and improve color purity, a blue homoleptic Ir(III) compound was created by incorporating a strong electron-withdrawing F atom into the *C*-coordinating phenyl of a dbi ligand [12]. As a result, the dbi-based Ir(III) compound was more efficient and longer-lasting than existing blue homoleptic Ir(III) compounds. Although there are many studies on iridium(III) compounds using C^N chelate dbi ligand and their results for optical properties, have been well established, reports on platinum(II) compounds using this ligand are scarce so far. This fact prompts us to synthesize dbi-based Pt(II) compounds and investigate their photophysical properties.

In this study, platinum was utilized as the metal core to investigate the effect on efficiency as compared to that of iridium congeners in phosphorescent OLEDS (PHOLEDs). A strongly electron-withdrawing fluorine atom was introduced onto the phenyl group of the main dbi ligand to widen the energy gap in the blue region. In addition, acetylacetonate (acac) and picolinic acid (pic) were added to investigate changes in the photophysical phenomena caused by the ancillary ligand [13,14]. Compounds **One** and **Two** were synthesized, and their structures were confirmed spectroscopically. The photophysical properties, including the crystal structure of **2**, were evaluated to assess their suitability as materials for organic lighting.

## 2. Materials and Methods

### 2.1. General Consideration

All experiments were carried out using the standard Schlenk method in a dry N_2_ atmosphere. According to a previous report [15], starting material, [PtMe_2_(μ-SMe_2_)]_2_, was synthesized. A detailed description of the instrumentation and reagents used is deposited in Appendix A.

### 2.2. Synthesis

Synthesis of [1-(2,4-diisopropyldibenzo-[b,d] furan-3-yl)-2-(4-florophenyl)-1*H*-imidazole-*N*,*C*^2′^)] platinum(II) (acetylacetonate-*O*,*O*) (**1**): To a 20 mL screw-cap vial was added one equivalent of fluorinated-dbi ligand (0.128 g, 0.310 mmol), [PtMe_2_(μ-SMe_2_)]_2_ (0.089 g, 0.155 mmol) and 4 mL of THF. The reaction was allowed to stir 1 h at room temperature, and then a solution of CF_3_SO_3_H (0.027 mL in THF (2 mL), 0.313 mmol) was slowly added. The mixture was stirred for an additional 30 min, and then a solution of Na(acac) (0.17 g in MeOH (3 mL), 1.21 mmol) was added. The mixture was stirred overnight and then partitioned between water and CH_2_Cl_2_. The organic layer was washed with water and dried over MgSO_4_. The residue was then purified by silica column chromatography (hexanes:CH_2_Cl_2_ as eluent, 1:1, *v*/*v*) to give analytically pure material. Yield: 0.1 g (46%). ^1^H NMR (400 MHz, CD_2_Cl_2_) δ 8.05 (d, *J* = 8.0 Hz, 1H), 7.95 (s, 1H), 7.64 (d, *J* = 8.1 Hz, 1H), 7.53 (t, *J* = 8.0 Hz, 1H), 7.41 (t, *J* = 8.1 Hz, 1H), 7.33 (d, *J* = 4.0 Hz, 1H) 7.13 (dd, *J* = 8.0, 2.0 Hz, 1H), 6.95 (d, *J* = 2.0 Hz, 1H), 6.28 (td, *J* = 8.0, 2.0 Hz, 1H), 5.5 (s, 1H), 2.62 (sept, *J* = 8.0 Hz, 1H), 2.55 (sept, *J* = 8.0 Hz, 1H), 1.96 (d, *J* = 8.0 Hz, 6H), 1.40 (d, *J* = 8.0 Hz, 3H), 1.26 (d, *J* = 8.0 Hz, 3H) 1.19 (d, *J* = 8.0 Hz, 3H), 1.09 (d, *J* = 8.0 Hz, 3H). ^13^C NMR (100 MHz, CD_2_Cl_2_) δ 185.6, 183.7, 157.0, 153.0, 141.2, 131.5, 130.4, 130.1, 128.1, 126.7, 124.6, 123.3, 123.2, 123.1(9), 123.1(4), 121.0, 120.9, 116.9, 116.7, 116.4, 112.0, 109.4, 109.2, 102.3, 28.7, 28.4, 27.7, 27.0, 25.1, 23.4, 21.6, 20.6. Anal. calcd for C_32_H_131_FN_2_O_3_Pt; C, 54.46; H, 4.43; N, 3.97; found: C, 54.50, H 4.45, N 3.92%.

Synthesis of [1-(2,4-diisopropyldibenzo-[b,d] furan-3-yl)-2-(4-florophenyl)-1H- imidazole-*N*,*C*^2′^)] platinum(II) (2-picolinate-*N*,*O*) (**2**): To a 30 mL screw cap vial was added one equivalent of fluorinated-dbi ligand (0.143 g, 0.348 mmol), [PtMe_2_(μ-SMe_2_)]_2_ (0.10 g, 0.17 mmol) and 15 mL of THF. The reaction was allowed to stir 2 hr at room temperature, and then a solution of 2-picolinic acid (0.653 g, 0.52 mmol in THF 10 mL) was slowly added. The mixture was stirred for an additional 48 h, then the addition of water and CH_2_Cl_2_. The organic layer was washed with water or brine and dried over MgSO_4_. The residue was then purified by silica column chromatography (EtOAC:CH_2_Cl_2_ as eluent, 1:3, *v*/*v*) to give analytically pure material. Yield: 0.08 g (32%). ^1^H NMR (400 MHz, CD_2_Cl_2_) δ 9.10 (d, *J* = 8.0 Hz, 1H), 8.18 ~8.5 (m, 3H), 7.95 (s, 1H), 7.65 (d, *J* = 15.0 Hz, 2H), 7.55 (t, *J* = 7.0 Hz, 2H), 7.42 (t, *J* = 10.0 Hz, 1H), 7.14 (dd, *J* = 8.0, 4.0 Hz, 1H), 6.99 (d, *J* = 4.0 Hz, 1H), 6.40 (td, *J* = 10.0, 2.0 Hz, 1H), 6.07 (td, *J* = 8.0, 2.0 Hz, 1H), 2.64 (quint, *J* = 6.0 Hz, 1H), 2.56 (sept, *J* = 6.0 Hz, 1H), 1.43 (d, *J* = 8.0 Hz, 3H), 1.28 (d, *J* = 8.0 Hz, 3H), 1.21 (d, *J* = 8.0 Hz, 3H), 1.10 (d, *J* = 8.0 Hz, 3H). ^13^C NMR (100 MHz, CD_2_Cl_2_) δ 157.0, 153.1, 148.5, 141.1, 139.3, 130.3, 129.7, 128.3, 128.2(7), 128.2(1), 128.1, 126.9, 126.6, 124.1, 124.0, 123.4, 123.30, 123.2(6), 123,2(1), 122.0, 121.9, 121.0, 119.1, 116.6, 112.0, 81.9, 28.8, 28.4, 25.1, 23.5, 21.6, 20.6. Anal. calcd for C_33_H_28_FN_3_O_3_Pt; C, 54.39; H, 3.87; N, 5.77; found: C, 54.38, H 3.82, N 5.75%.

### 2.3. Single Crystal X-ray Analysis

X-ray diffraction data for Complex **Two** were collected at 173 K on a Bruker SMART APEX II ULTRA (Bruker AXS GmbH, Karlsruhe, Germany) diffractometer. Using the software package of APEX2 [16], data collection/reduction and semi-empirical absorption correction (SADABS) [17] were conducted. All of the calculations for the structure determination were carried out using the APEX2 package with the SHELXS-2014 (Version 6.22) [18] and SHELXL-2014 (Version 6.22) [19] programs. The non-hydrogen atoms of Compound **Two** were refined anisotropically. All hydrogen atoms were added in calculated positions and refined isotropically in a riding manner. The crystal structure figure was made using the Diamond program (Version 3.2) [20]. A summary of the refinement details and resulting factors for the crystal structures of **Two** are given in Appendix A. CCDC number **Two** is as follows: CCDC-2259034.

### 2.4. Device Fabrication and Measurement

The PHOLEDs structure is as follows:Mixed host: mCBP/CNmCBPCN;Dopant and doping ratio: Compound **One** at 3 and 10 weight %.

Detailed device structures, including the common layer and the full name of corresponding compounds, are presented in the Appendix A. Device experimental conditions and measurements are the same as those reported in our previous work [21].

## 3. Results and Discussion

### 3.1. Synthesis and Structure

Platinum complexes **1** and **2**, incorporating the main **dbi** ligand, were synthesized according to a previously reported procedure [22], as shown in Figure 1. Yields of 32–46% were obtained by treating the **dbi** ligand with [Pt(μ-SMe_2_)Me_2_]_2_ in THF and then adding trifluoromethanesulfonic acid (TfOH). Either Na(acac) or 2-picolinic acid was subsequently added to the reaction mixture. Both complexes were characterized using various spectroscopic methods, and the crystal structure of **2** was determined.

The crystal structure and selected bond lengths (Å)/angles (°) of **2** are shown in Figure 1. Compound **2** had a square planar geometry around the Pt ion with bite angles ranging from 80° to 174°. The Pt–C, Pt–N, and Pt–O bond lengths in Compound **2** were consistent with those of similar Pt(II) compounds in previous studies [23,24].

Interestingly, various intermolecular interactions existed in the packing structure, such as H–C(π) and C(π)[dbi]–C(π)[pic], as shown in Figure 2. However, the typical Pt-to-Pt interaction expected in square-planar Pt(II) compounds was not observed, which may be due to the high steric hindrance of the ^i^Pr unit in the ligand. The crystal data and refinement structure of **2** are listed in Appendix A.

### 3.2. Photophysical and Electrochemical Properties

Both absorption and emission spectra of **1** and **2** at 298 K and 77 K are shown in Figure 3. Both compounds exhibited intense absorption (extinction coefficients (ε) = ca. 2.0–4.6 × 10^4^ M^−1^ cm^−1^) from approximately 230 to 280 nm. The first intense band was attributed to a ligand-centred (LC) π–π* transition, while the second moderately intense band may be due to a metal-to-ligand charge-transfer (MLCT) transition [25]. In addition, a weak absorption band originating from the triplet transition was observed in the region of 460–480 nm (See Appendix A). Although **1** and **2** exhibited differences in their absorption intensities, the absorption patterns of both complexes were similar. However, the absorption wavelengths and molar extinction coefficients differed between the complexes. For example, Complex **1** exhibits ^1^MLCT at around 326 nm with a reasonable molecular extinction coefficient (See Table 1), while for Complex **2**, red shifted absorption (ca. 333 nm) with ε = 0.63 × 10^4^ [M]^−1^cm^−1^ appeared in the ^1^MLCT region.

The emission spectra of **1** and **2** measured at 77 K and 298 K in THF are shown in Figure 3. Both compounds exhibited bright bluish-green emissions with similar emission maxima (ca. 480 nm) in the THF solution. The 77 K emission spectra of both compounds indicated that the replacement of acac with pic resulted in a red-shifted triplet energy. The greater triplet energy of **1** compared with that of **2** could be due to the deeper HOMO energy levels of the ancillary ligand because the electronic transition of **1** primarily originates from the ancillary ligand rather than the main C^N ligand. In contrast, the pic ancillary ligand contributed little to the HOMO level. This trend is in agreement with the density functional theory (DFT) calculations (see Section 3.3). Interestingly, the emission spectrum of **2** at room temperature exhibited an additional band at a shorter wavelength of 400–450 nm with a similar decay time (τ: 30 µs) to that of the emission maximum at 482 nm (Appendix A). The similar decay times at both the additional emission (ca. 430 nm) and emission maximum may indicate a metal-assisted delayed fluorescence (MADF; T_1_ → S_1_ → S_0_). A similar trend was recently reported for Cu(II) and Pd(II) complexes [26,27,28]. However, to confirm this result, further studies, such as the measurement of excitation, thermally activated emission, and the actual energy gap between the S_1_ and T_1_ states, are necessary.

The triplet energies (E_T_) of **1** and **2** were obtained from the onset of the phosphorescent emission spectra of mCP-doped thin films and were estimated to be 2.74 and 2.61 eV, respectively. These values are comparable to those of the homoleptic Ir(III) analogs, Ir(dbi)_3_ (E_T_ = 2.58 eV) and Ir(F-dbi)_3_ (E_T_ = 2.71 eV) [12]. However, they are very close to those of Pt(II) β-diketonate derivatives, Pt(C^N)(O^O) (E_T_ = 2.72~2.73 eV), where C^N is a bipyridine ligand [29].

The absolute photoluminescent quantum efficiencies (Φ_PL_), measured using an integrating sphere, were 0.37 for **1** and 0.28 for **2**. The efficiencies of both complexes were poorer than those of the similar Ir(II) complexes Ir(dbi)_3_ (Φ_PL_ = 0.52) and Ir(F-dbi)_3_ (Φ_PL_ = 0.50) and even Pt(C^N)(O^O) (Φ_PL_ = 0.60~0.65). The molecular rigidity through both intramolecular interactions and the number of intermolecular interactions between two adjacent molecules in packing is closely related to the resulting PLQY [30,31,32]. Therefore, the low PLQY of **1** and **2** may be attributed to the weakened molecular rigidity due to the ancillary ligands, as indicated by the crystal packing. Therefore, heteroleptic Pt(II) compounds have disadvantages in terms of phosphorescent quantum efficiency compared to homoleptic Ir(III) analogs. Consequently, Complex **1**, bearing acac as the ancillary ligand, exhibited a greater triplet energy and photoluminescence quantum yield (PLQY) than Complex **2**, which has a pic ancillary ligand. The photophysical properties of **1** and **2** are presented in Table 1.

To investigate the effect of the ancillary ligand on oxidation, cyclic voltammetry experiments were conducted on both compounds. A quasi-reversible oxidation at E_OX_ (onset) (vs. FeCp_2_/FeCp_2_^+^) = 0.93 V and 1.03 V was observed for both compounds, as shown in Appendix A. The oxidation potential of **1** was slightly lower than that of **2**, which agreed with a previous report on Ir(III) compounds containing either acac or pic ancillary ligands [33]. The C^N chelate ligand formed a stronger bond with the Pt(II) ion in **2** than with that in **1**. This plays a key role in increasing the HOMO level because the *p*-orbitals of the C^N chelate and the *d*-orbital of Pt(II) are the main contributors to the HOMO level (see Section 3.3). Moreover, the results indicate that acac acts as a weak trans-directing ligand compared to picolinate.

The HOMO energies for **1** and **2** were estimated to be −5.73 eV and −5.83 eV, respectively, based on the energy level of Cp_2_Fe (4.8 eV below the vacuum level) [34]. By combining the HOMO energy and the optical band gap, the LUMO energies were determined to be −2.64 eV for **1** and −2.69 eV for **2**. The electrochemical properties and energy levels of **1** and **2** are presented in Table 2.

### 3.3. TD-DFT Calculations

To understand the electronic transitions and emission properties of both compounds, molecular orbital calculations were conducted using time-dependent density functional theory (TD-DFT). The geometrical optimization of **1** was performed in the vapor phase, whereas the optimized geometry of **2** was obtained from its crystal structure [35]. Based on the calculation results, the S_0_ → S_1_ transitions mostly originated from the HOMO and LUMO with acceptable oscillator strengths (Table 2 and Appendix A). Henceforth, the discussion of the calculation will primarily focus on the HOMO and LUMO. Although the calculated values for both the HOMO/LUMO energies and energy gaps in **1** and **2** were significantly different from the measured values, the trends in the calculated and measured values were in reasonably good agreement. As shown in Figure 4, the HOMO levels of **1** and **2** had significant contributions from the Pt (*d*-orbital) ion and major contributions from the π-orbitals of the C^N chelate F-dbi ligand. However, the contribution from the 5*d*-orbital was significantly different in the HOMO level of the complexes, namely 36% for **1** and 43% for **2**. For compound 1, the LUMO distribution was predominantly localized on the π*-orbitals of the dibenzofuranyl unit of the C^N chelate, with a negligible contribution from the acac ancillary ligand. In contrast, compound 2 exhibited a significant contribution from the pic ancillary ligand and a minimal contribution from the C^N chelate in the LUMO level. Notably, unlike the HOMO levels, the *d*-orbital contributions to the LUMOs of both complexes were negligible, ranging from 0% to 2%. The values of the S_0_ → T_1_ transitions of **1** and **2** were estimated to be 422.5 and 443.8 nm, respectively, which was in agreement with the onset of the experimentally measured ^3^MLCT in the absorption spectra (Appendix A). Consequently, our calculations suggest that the ancillary ligand plays a key role in determining the electronic transitions, including the degree of MLCT. Based on the calculations, the electronic transitions in **1** might arise from a ligand-centered (LC) charge transfer (π_dbi_–π*_dbi_) combined with MLCT (Pt*_d_*–π*_dbi_), whereas for **2**, a ligand-to-ligand charge transfer (LLCT; π_dbi_–π*_pic_) combined with MLCT (Pt*_d_*–π*_pic_) might be considered as a key electronic transition.

### 3.4. PHOLEDs Performances

An attempt was made to fabricate a device using **1** and **2** as dopants. However, compound **2** decomposed during the vacuum purification process, resulting in impure samples that were not suitable for device application. Therefore, only device data for compound **1** were obtained, and the following discussion will focus primarily on compound **1-**based devices with different doping levels. A device structure confining triplet exciton was used to evaluate compound **1** by employing a mixed host of mCBP:CNmCBPCN (50:50). The current density-voltage and luminance-voltage data in Figure 5a shows the current density and luminance change according to the applied voltage. A decrease in current density was noticed at a high doping concentration by hole trapping originating from the HOMO gap between the p-type mCBP host (−6.00 eV) and compound **1**. However, the driving voltage of the devices at 1000 cd/m^2^ was 5.9 V, both at 3% and 10% doping concentrations. In spite of low current density at 10% doping concentration, high EQE led to similar luminance at the same driving voltage. The external quantum efficiency (EQE)-luminance data in Figure 5b present a maximum EQE of 10.4% at 10% doping concentration. The EQE was relatively low at 3% doping concentration because of imperfect energy transfer from the host to compound **1**, as identified from the electroluminescence (EL) spectra in Figure 5c. An additional pick with low intensity appeared at around 420 nm, which might be attributable to the host [33]. The maximum current efficiency of the compound **1** device was 18.0 cd/A at 10% doping concentration.

Two main emission peaks were delivered in the EL spectra of compound **1**, and they were observed at 460 nm and 490 nm. The EL spectra of compound **1** were similar to the PL spectrum, although the relative intensity of the two peaks was slightly different by the cavity effect in the devices. The color coordinates of the compound **1** devices were (0.187, 0.316) at 3% doping concentration and (0.187, 0.335) at 10% doping concentration. To compare with previous work, an additional PHOLEDs device with the same structure was also fabricated with Pt(tpim)(O^O) as a reference, where tpim is a 1-([1,1′:3′,1″-terphenyl]-2-yl)-2-(4-fluorophenyl)-1*H*-imidazole [24]. The external quantum efficiency of **1** based device at a 10% doping level is lower than that of standard Pt(tpim)(O^O) under the same luminance, which could be due to the incomplete energy transfer and low PLQY. Summary regarding electroluminescence characteristics of **1** and reference Pt(II) material, Pt(tpim)(O^O), are deposited in Appendix A.

## 4. Conclusions

We synthesized two new platinum compounds based on a C^N chelated dbi ligand and systematically investigated the effect of different ancillary ligands, acac (**1**) and pic (**2**), on the photochemical properties of Pt(II) compounds. X-ray diffraction analysis of **2** revealed a square-planar structure and various intermolecular interactions in the packing structure. Although there was no significant change in the maximum emission wavelength (480 nm) upon changing the auxiliary ligand, **2** exhibited the expected MADF peak with relatively strong intensity in the short-wavelength region (400–430 nm). Furthermore, the incorporation of acac as an auxiliary ligand led to an increase in the quantum efficiency and triplet energy compared to those of the pic ancillary ligand. TD-DFT calculations showed that the electronic transition for **1** might arise from a ligand(C^N)-centered (LC) π–π* transition combined with an MLCT transition. In contrast, the main contributions in **2** originated from an MLCT combined with a ligand(dbi)-to-ligand(pic) charge transfer (LLCT) due to the minimal contribution from the C^N chelate of the fluorinated-dbi to the LUMO levels. We successfully fabricated multilayer OLED devices using an mCBP/CNmCBPCN mixed host with **1** as the dopant. The resulting PHOLED device exhibited a bright sky-blue emission, with a maximum EQE of approximately 10.4% and a current efficiency of 18.0 cdA^−1^. This study highlights the importance of the ancillary ligand in the design of blue phosphorescent Pt(II) compounds and can help to expand the use of organic lighting in energy-efficient applications. Further investigations on the effect of ancillary ligands on the performance of Pt(II) complex-based OLEDs are underway in our laboratory.

## Data Availability

Data generated or analyzed during this study are provided in full within the published article and its Appendix A.

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
