# Peer review of "Development of Blue Phosphorescent Pt(II) Materials Using Dibenzofuranyl Imidazole Ligands and Their Application in Organic Light-Emitting Diodes"

_materials, 2023, doi:10.3390/ma16114159_

Round 1

Reviewer 1 Report

Reviewer comments on Manuscript Development of Blue Phosphorescent Pt(II) Materials using Dibenzofuranyl Imidazole Ligands and their Application in Organic Light Emitting Diodes 

The authors presented two sky-blue phosphorescent Pt(II) compounds with a C^N chelate, fluorinated-dbi (dbi = [1-(2,4-diisopropyldibenzo [b,d]furan- 3- yl)-2-phenyl-1H-imidazole]), and acetylactonate (acac)/picolinate (pic) ancillary ligands. One of the ligands was used as an emitter in PhOLED with it’s EQE reaching 8.4 %. Experimental results were substantiated by theoretical calculations. The overall impression of the research article is good. Although, I would recommend this article to be published in Materials after major revision, as some more experiments would be useful for the research article:  

  • Additional theoretical calculations are nescessary to make the experimental results more in line. Different functionals should be tested for the geometry optimization as well as the media polarity should be taken into account. 

  • It was stated that OLED based on compound 2 as emitter was not fabricated because the compound had decomposed during the vacuum deposition. It would be of great value to fabricate the device by solution process as well as that based on compound 1, as it could be compared more essentially. 

  • The comparison of OLED characteristics determined for the fabricated devices should be compared with the PhOLEDs already described in the literature, such comparison would enhance the overall device part description and help compare the presented compounds with other emitters that are already presented.

The overall English is good.

Reviewer 2 Report

The authors study the properties of Pt complex by spectroscopy and theory method. For molecule 1, the multilayered phosphorescent OLED was successfully fabricated, and the performance is investigated. For this manuscript, I would suggest publishing after a minor revision as following:                                 

            Comparing the Pt complex in this manuscript with Ir(III) compounds, authors mentioned Pt complex will achieve greater efficiency (Page 2).   However, the PL efficiency of Pt complexes are poorer than Ir (Page 6). These statements are contradictory. How to demonstrate the advantages of Pt complexes.

            In Table1, what is the meaning of the superscript a,b,c ? and how to get the absorption/emission properties: in solution or spin-coating film or doping film?  I found in the main text, the emission spectra are from THF solution, which is in CH2Cl2 in Figure 3. The absorption spectra shown in Figure 3 seems only 298 K. What is the lifetime measurement condition: temperature, detecting wavelength?

            For PhOLEDs performance of molecule 1, the doping concentration are shown here are 3% and 10%. 10% doping is the best or just better than 3%? Did authors test other concentrations? And there is an additional peak ~420 nm for 3% device, not sure what is it from.

            Based on the calculations, the transitions are different for 1 (pi-pi* + MLCT) and 2 (MLCT + LLCT). However, from the absorption spectra description, the two peaks are from pi-pi* and MLCT for two complex. The oscillator strength of S0àS1 from calculations seems a little low for an allowed transition.

            From this manuscript, no crystal structure for complex 1.  Is it hard to get the crystalline phase?

Reviewer 3 Report

Lee, Kang and collaborators described the synthesis and characterization of two platinum organometallic compounds for their potential application in OLEDs. In general, the work presented here was designed property, however, before the acceptance of this work in Materials, I would like to recommend some changes that need to be addressed in order to improve this work. 

1. Authors made a good work in the introduction for this manuscript, and they mentioned at least four different complexes with Iridium (III) as metal center. However, this is not the first work reporting Platinum (II) complexes for light-emitting devices and most important, in the literature it is possible to find some examples of Pt complexes containing acetylacetonate and picolinate as ligands. Considering the above, author should include some examples of Pt complexes to have a clear idea of the impact of this work. 

2. The description of the synthetic procedures needs to be reported in past tense. 

3. In section 3.1 authors only indicated the addition of Na(acac) but they should mention also the pinacolic acid. 

4. In section 3.2 author indicated that a weak transition band is observed between 460-480 nm in Figure 3, however, there are not any peak in that region. Please verify. In addition, it would be important to know how authors justify the absence of MLCT bands in complex 1, is it related to the difference in the emission peaks at room temperature for both compounds? Certainly, they are different despite authors said that they are similar.  

5. To make easier the analysis of crystal structure of compound 2, authors could merge figures 1 and 2. 

6. The triplet energies obtained experimentally for compounds 1 and 2 are compared in the main text with the energies for Ir(dbi)3. However, a strict comparison should be done using the analogous compound with Pt(II). Authors are claiming that platinum complexes have greater efficiency than iridium complexes, so they should make all experimental comparisons between platinum complexes.

7. Authors mentioned that the poor PL efficiencies in compounds 1 and 2 may be attributed to weakened molecular rigidity as indicated by the crystal packing. To argue this, author could analyze the thermal ellipsoids in the crystal structure of compound 2 and verify if there is any suggestion of molecular motion in the crystal. Is there crystallographic disorder in the structure? Any other proof to support this argument of less rigidity in 1 and 2 than iridium complexes in reference 29? 

Round 2

Reviewer 1 Report

The authors have not sufficiently responded to the comments neither have modified the manuscript. For example, the last comment regarding the comparison of their PhOLED results with those of already published in the literature, the addition of the table is not sufficient. A comment in the main text should be added.

The authors have declined to fabricate a device by solution-proccess by enhancing the nescessity of pure compound. Such statement raises a doubt whether the presented new compounds have been sufficiently purified and whether their results are of pure compounds or those with impurities. I'd like to ask to present HPLC results of the presented compounds to confirm the purity of the compounds.

The overall English is good.

Round 3

Reviewer 1 Report

I acknowledge that the authors have taken into account my comments, and suggest to accept this version of the manuscript.

English level is sufficient.